# Barriers and Facilitators to the Learning and Acquisition of Research Competencies among Nursing Students through Active Methodologies: A Qualitative Study Using Reflective Writing

**DOI:** 10.3390/healthcare11081078

**Published:** 2023-04-10

**Authors:** Juan Francisco Velarde-García, Beatriz Álvarez-Embarba, María Nieves Moro-Tejedor, Leyre Rodríguez-Leal, Oscar Arrogante, María Gema Alvarado-Zambrano, Jorge Pérez-Corrales, Domingo Palacios-Ceña

**Affiliations:** 1Department of Nursing, Red Cross College of Nursing, Universidad Autónoma de Madrid, Avenida Reina Victoria 28, 28003 Madrid, Spain; 2Research Nursing Group of Instituto de Investigación Sanitaria Gregorio Marañón (IiSGM), Calle del Dr. Esquerdo, 46, 28007 Madrid, Spain; 3Research Group of Humanities and Qualitative Research in Health Science (Hum & QRinHS), Universidad Rey Juan Carlos, Avenida Atenas s/n, 28922 Alcorcón, Spain; 4Faculty of Nursing, Physiotherapy and Podology, Complutense University of Madrid, Plaza de Ramón y Cajal, 3, 28040 Madrid, Spain; 5Hospital Universitario La Paz, Paseo de la Castellana, 261, 28046 Madrid, Spain; 6Department of Physical Therapy, Occupational Therapy, Physical Medicine and Rehabilitation, Universidad Rey Juan Carlos, Avenida Atenas s/n, 28922 Alcorcón, Spain

**Keywords:** project-based learning, small-group learning, self-directed learning, nursing students, qualitative research

## Abstract

Background: The development of educational research, critical thinking skills, and evidence-based practice requires proposals for educational innovation. The purpose of this study was to explore the perspectives of undergraduate nursing students on the barriers and facilitators after the implementation of a novel activity within the course of research methodology, composed of three active learning strategies: (a) project-based learning; (b) small-group learning; and (c) self-directed learning. Methods: A qualitative exploratory study using reflective writing was conducted at the Nursing Department of the Red Cross School (Spain). Seventy-four nursing students participated in the study, enrolled in the research methodology course. Purposive sampling was used. Online reflective notes were collected from a script of open-ended questions. An inductive thematic analysis was performed. Results: The new proposals facilitated learning of the subject matter and its contents. They were useful and enabled the students to put the contents into practice. In addition, they improved the students’ organization, planning, and involvement. The barriers identified were a lack of time, ambiguity, inadequate tutoring or novelty of the work, and inequity in the distribution of tasks and workloads. Conclusions: Our findings shed light on the barriers and facilitators identified by nursing students when implementing an educational innovation proposal, using three active learning methodologies as learning tools for the subject of nursing research.

## 1. Introduction

Research activities are considered high-impact educational practices, as they enable lifelong learning of skills and attitudes [1,2]. Undergraduate research was defined as any teaching and learning activity where students are actively involved with the content, process, or problems of research in their discipline. Specifically, Healey described a framework including examples in which undergraduate students are actively engaged with the research content, or processes and problems, of their discipline, as well as models in which students act as a more passive audience, receiving information about the research content, or processes and problems of their discipline [3].

The objectives of nursing education institutions include providing future nurses with the competencies necessary to provide high quality nursing care [4]. Students are expected to learn how to use the knowledge provided by research and to acquire skills, competencies, and behaviors in their daily clinical practice [4]. Therefore, competency-based education (CBE) has become the mainstay of practice and research in many training programs [5,6,7]. Its use makes it possible to establish measurable parameters for researchers and to determine the effectiveness of such programs. In turn, competencies allow programs to be flexible to accommodate time constraints, as well as the needs of individual learners [8,9]. CBE provides students the opportunity to learn according to their needs, so that they can apply the acquired knowledge, skills, and abilities to the current and future challenges in their work environment [10,11,12].

To acquire research competencies, the nursing curricula must meet the demands of the real world by teaching and evaluating the available evidence to help create a culture that uses evidence to update clinical practice [13]. The term “evidence-informed practice” highlights the need to recognize and address contextual influences and consider how the best available evidence can be used in specific circumstances [14]. Undergraduate nursing students must learn to translate knowledge, skills, competencies, beliefs, attitudes, and behaviors into daily practice, and improve clinical outcomes [15]. Knowing how to use the best available evidence enables patient care decisions to be made by integrating the most current and valid research findings, the nurse’s clinical experience and values, patient preferences, and available resources. This leads to an improvement in practical co-knowledge and patient care, in addition to reducing adverse events and patient care costs [16].

Within CBE, a variety of teaching–learning strategies applicable to undergraduate nursing students have been employed that enable them to gain confidence in accurately interpreting literature and research [17]. Moreover, there is a need to seek models that support the integration of research competencies beyond the classroom [18]. Higher education is moving from a teacher-centered teaching model to a student-centered learning model. Based on this, experiential learning theory supports the creation of knowledge through the transformation of experience. Experiential learning theory includes: (1) concrete experience, (2) abstract conceptualization, (3) reflective observation, and (4) active experimentation. Through experiential learning, the learner constructs knowledge based on experience, reflection, knowledge, and action [19].

At present, increasing relevance is given to student-centered teaching and learning strategies [20]. Thus, the teaching method used with students could become an obstacle, since some may achieve more superficial learning versus those who achieve more in-depth learning and thus a greater understanding of the facts [21]. Active learning has been shown to be effective for enhancing students’ learning processes [22,23]. Moreover, according to the guidelines of the European Higher Education Area, students are considered active learners and responsible for their own development and for the specific competencies acquired [24]. Active learning methods, such as flipped classrooms, problem-based, team-based or case-based learning, are learner-centered methods, developed for use beyond traditional large-group lectures [25].

The active teaching–learning strategies used include project-based learning (PBL), small-group learning (SGL), and self-directed learning (SDL). Through PBL, students are expected to apply knowledge, improve their skills, and obtain results by participating in the resolution of authentic real-world problems [26]. Students learn by gathering and constructing knowledge through the activities they develop. In addition, PBL generates high satisfaction and pleasure in students; together with greater proactive learning, performance of assigned functions and increased motivation [27]. SGL, on the other hand, is used in nursing education to foster individual inventive learning and collaborative attributes, as well as knowledge acquisition and retention [28]. It allows students to actively participate in their learning and enhances both personal and professional development [29]. Through this method, problem solving, time management, interpersonal communication, and critical thinking skills are developed, leading to lifelong learning [30]. Finally, SDL allows individuals to take the initiative, with or without the help of others, to identify their own learning needs, formulate learning goals, identify human and material resources for learning, choose and implement appropriate strategies, and subsequently evaluate their learning outcomes [31]. This strategy has been applied as part of learning models in nursing education [32]. In a study conducted in Turkey, SDL was significantly associated with academic performance [33]; whereas in Korea, SDL was associated with the communicative competence of nursing students [34]. Additionally, a systematic review on the application of SDL in the education of health professionals, including nursing and medicine, reported improved skills and attitudes [35].

Based on the advantages offered by the new teaching–learning strategies, a teaching innovation activity was carried out in the subject of research methodology within the nursing degree, integrating PBL, SGL, and SDL. Until then, the contents were taught based on lectures and final exams, according to the traditional models of education. This model is based on the reading of textbooks, the use of didactic material, the incorporation of objective tests and the critique of scientific articles. Traditional teaching is centered on the teacher, who uses blackboards, projectors, and images [36]. In contrast, this teaching innovation sought to introduce a novel aspect into the existing reality. Educational innovation implies the creation of new knowledge, products, and processes, as an essential part of the work in the 21st century societal organizations [37].

In addition, reflection is fundamental to learning in the clinical healthcare context [38]. Reflection is a self-dialogue that increases the individual’s awareness of their learning and clinical practice, allowing them to look back on an event and reflect on how they thought about and responded to it [39]. In nursing students, it helps to critically analyze their clinical experiences in their learning process [40], increasing critical thinking, awareness about their behaviors and decisions in their training [41]. Reflective writing improves learning outcomes and positively influences students’ progress [42]. The role of reflective writing in the clinical placements of nursing students has been described and analyzed; however, there are no studies on the learning and acquisition of competencies in nursing research [43]. In turn, for the researchers of this study, it was necessary to know whether the use of the teaching strategies employed within the subject enabled the acquisition of research competencies in terms of the knowledge, skills, and abilities expected in an undergraduate student.

Therefore, the aim was to explore the perspectives of undergraduate nursing students on the barriers and facilitators of implementing a teaching innovation activity using three active learning methodologies to acquire knowledge and competencies in the nursing research process.

## 2. Materials and Methods

### 2.1. Design

A qualitative exploratory study using reflective writing was conducted, based on an interpretive framework [44]. This study was conducted according to the Consolidated Criteria for Reporting Qualitative Research [45] and the Standards for Reporting Qualitative Research [46].

### 2.2. Context and Setting

This study was conducted at the Nursing School of the Spanish Red Cross in Madrid (NSSRCM). In Spain, the nursing degree consists of 240 ECTS credits (6000 h), completed across four university years (European Credit Transfer System), and is included in the European Higher Education Area [47]. Among the competencies to be developed by students, research is the common thread for learning nursing interventions in evidence-based practice, and applying research methodology at the level of care, teaching, or management [48].

In the study plan, research competencies are developed in the third-year subject “research methodology”. Among the learning outcomes, students are expected to apply and justify the stages of a research project, and perform literature searches and critical readings related to a research question through the elaboration of a research project [48]. To this end, a teaching innovation activity was developed using three active learning methodologies: (1) PBL, (2) SGL, and (3) SDL. (Figure 1).

Prior to the use of these methodologies, students’ learning derived exclusively from lectures, as the only means of teaching used by the teachers, there was very little interaction, only complementary documentation was provided to the classes, and there was no record of the level of knowledge and skills acquired during the delivery of the course. After teaching the subject, the students were tested and graded based on a final exam.

### 2.3. Research Team and Reflexivity

Prior to the study, the researchers’ positioning was established via briefing sessions addressing the theoretical framework for this qualitative study, their beliefs, and their motivation for the research [45]. The authors of this study departed from the assumption that the contents taught within the subject of research methodology are very different from those received until then: students have little previous experience throughout their education and may not find the subject matter very attractive, unlike other subjects. For this reason, an activity was proposed using different strategies to stimulate and motivate students’ learning. Moreover, the learning strategies were already used in other subjects in previous courses, and therefore the students were familiar with them.

The research team was formed by eight researchers (four women), including seven nurses and one occupational therapist (JPC). Three researchers had experience in qualitative study designs (DPC, JPC, JFVG). Three of the researchers were in charge of teaching the subject and had teaching training provided by the University, in addition to previous experience with active teaching and learning methodologies (JFVG, LRL, MNMT), specifically. The main researcher and coordinator of the subject (JFVG) was qualified as an expert in university teaching.

### 2.4. Inclusion/Exclusion Criteria and Sampling Strategies

A purposeful sampling strategy was employed [49]. The inclusion criteria consisted of nursing students at the NSSRCM, who were students in their third year of undergraduate studies, enrolled for the first time in the “research methodology” subject. No students enrolled in the course were excluded.

During the 2021–2022 academic year, of the 92 students enrolled in the subject, 78 participated in the study. A sampling of “richness and volume of data, and pragmatic considerations” were applied [50]. Considering the first criterion, the entire university under study (enrolled students) was included, obtaining a greater richness of data. Regarding the second criterion, the pragmatic criterion used was the possibility of accessing and recruiting the participants, as they were all enrolled in the same subject [50]. There were no dropouts.

### 2.5. Data Collection

Data were collected from online reflective notes obtained from an open-ended question guide, where students reflected and wrote about their perspective on the learning process used for acquiring research competencies [49]. The reflexive questions covered the three methods used (Table 1). Data collection through the questions lacked prior piloting and the participants were only asked the questions once. The reflective notes provided written texts; therefore, it was not necessary to record or transcribe the collected data, and field notes were not collected.

Within the different forms of knowledge as a framework for developing reflective practice among nursing students [51], in the present study, Baker’s four-step model was used [52], which describes the reflective process of writing based on the following steps: (1) identification (selecting an experience during your learning that stands out significantly); (2) description (detailing thoughts, feelings, and events of the experience); (3) significance (drawing personal meaning from the experience); and (4) implication (explaining how the experience affected you) [52].

### 2.6. Analysis

A thematic, inductive analysis was performed. The thematic analysis consisted of identifying the most descriptive content to obtain codes, and subsequently reduce and identify the most coded groups (categories). Accordingly, categories were formed [46]. This thematic analysis process was performed separately on each reflective note. Subsequently, joint meetings were held to combine the results of the analysis and to represent the participants’ perspective [46]. In cases of potential discrepancies, theme identification was based on establishing a consensus between the research team members. During the analysis, data saturation was apparent, as the data were repeated, and no new information was provided. A qualitative software was not used to analyze the data.

### 2.7. Ethics

Our study received the approval of the corresponding Research Ethics Committee of NSSRCM (approval code JVG_EUCREU 1/22). In addition, it was evaluated and supervised by the university’s teacher training department and the research committee. The anonymity and confidentiality of all participants was maintained at all times. Confidentiality of the results was ensured by omitting the students’ personal data and assigning codes consisting of numbers and letters, thus preventing their identification. Informed consent was obtained from all participants. The study followed the principles and ethical guidelines for medical research with human subjects adopted by the Helsinki Declaration.

Students’ participation in the study was voluntary and participants were given the opportunity to withdraw from the study at any time. The professors encouraged the students to participate by highlighting the benefits in terms of the acquisition of knowledge and research skills in the elaboration of the final project and its usefulness as future professionals. They also highlighted the learning difficulties identified by students in previous years.

### 2.8. Rigor

Several techniques were used in this study [49]: to ensure reliability, researcher triangulation was used, where data analysis was performed by several researchers. Subsequently, team meetings were held to compare the analysis and identify the main themes. In this manner, data credibility was ensured. Transferability was established by providing details of the investigators, participants, sampling strategies, data collection procedures, and data analysis. Reliability was achieved with an external auditor, who evaluated the research protocol, focusing on aspects related to the study methods and design. Finally, confirmability was attained by applying researcher reflexivity.

## 3. Results

Seventy-four nursing students (71 women) participated. The mean age was 21.4 (SD 1.75) years. Barriers and facilitators for learning research competencies in relation to PBS, SGL, and SDL were described (Figure 2).

### 3.1. Project-Based Learning

#### 3.1.1. Facilitators

In the reflections, the students acknowledged that PBS enhanced their learning process and the integration of the subject contents: “it has been a great help to integrate knowledge and understand it, not simply memorizing information but giving meaning and sense to what is being studied.” (E39). The students believed that PBL served to consolidate the subject matter: “the activities were based on the syllabus, therefore, it helped me to understand it.” (E45).

Students’ reflective texts highlighted the usefulness of PBL by helping them take a more active role in their learning and allowing them to put theory into practice: “it’ s useful because we put everything we learned into practice, and it wasn’t just theory.” (E4). They found it useful because it also helped them synthesize the contents received: “… it’s quite useful for summarizing and remembering the contents of the course.” (E50).

PBL facilitates study, when it comes to assimilating concepts: “doing this work helps to assimilate many concepts, facilitating the later study of the subject.” (E59). In addition, in their reflections the students pointed out how PBL helped them prepare for exams, because they were obliged to study the contents as they carried out the projects: “to do the project you had to know the syllabus, so you have already done part of the work for the exam.” (E63).

One student, in her reflections, felt that PBL was the only useful source of knowledge: “I believe that the knowledge I have of this subject is solely based on my work for the project.” (E52).

#### 3.1.2. Barriers

The students’ reflective texts showed that PBL was not useful when they lacked the time to learn it, minimizing what they had learned: “it’ s done so quickly and in such a short time that you learn the bare minimum.” (E72).

In addition, the students highlighted a lack of understanding of the activity due to the research content taught: “often, even the syllabus itself was hard to understand, it’s a difficult subject.” (E57). The students claimed that the contents gave rise to different interpretations, generating ambiguity and confusion among the students: “… at the beginning it was a bit ambiguous” (E19); “It has been more cumbersome until we have come to understand the procedure.” (E46).

Another barrier pointed out was the scarce guidance provided by the teachers during the development of the activity: “the difficulties have sometimes been the lack of guidance and guidelines from teachers.” (E33). Several students reflected on the difficulties encountered due to the novelty of the methodology and because it was the first time they used it: “I found it very difficult to carry out the project because it was something we had never done before.” (E50).

### 3.2. Small-Group Learning

#### 3.2.1. Facilitators

The students described that working in small groups was more equitable for the distribution of tasks and there was greater involvement of their peers: “it’s better this way, we all work the same amount and are 100% involved.” (E8). Another advantage was that the contributions of each member were better known, as opposed to larger groups: “in such small groups one can tell who has done the work and who hasn’t, in larger groups participation is masked.” (E18).

In addition, small groups facilitated the organization of tasks and interaction among its members. The students reflected that the distribution of tasks was more orderly: “working in smaller groups has been a great help, you reach an agreement earlier, and you distribute the tasks in a more orderly way.” (E67).

In their reflections, the students described how the small groups facilitated communication and participation among their members: “the smaller number of members in the group means there is more communication, and all members can participate in each part.” (E40). In turn, they facilitated individual contributions within the project: “it’ s a way for everyone to bring their expertise to the project” (E50). It also helped share the workload: “brainstorming is much easier when the working group is smaller.” (E73).

Several students acknowledged that in small groups, there is a greater knowledge of the work done, as it is distributed among fewer members: “you have a better understanding of why each part of the work is done, if there are more of us you end up dividing it up and only develop your own work.” (E64). In these circumstances, students feel “compelled” to do their assignment and to know the subject matter in order to undertake the coursework: “it forces you to do your part, especially for those who don’t come to class, it forces them to know what the subject is about because otherwise they can’t work.” (E54).

For the students, these groups generated greater satisfaction, fostering a good working environment and student contributions, and the result is of greater quality compared to work with more members: “we’ve managed to organize ourselves really well, we have a good working environment, and we have all contributed... I feel that it would be impossible to carry out this work with more people without losing quality.” (E24).

#### 3.2.2. Barriers

Workload and dedication required outside of class hours were cited as barriers: “I would have liked to have more time in the classroom to spend my afternoons studying other subjects.” (E26). Another barrier was the great dedication and extensive work involved (time investment), which was perceived as not very useful: “There isn’t enough time to carry out the activity; it requires more time and dedication than we have available.” (E70).

A further barrier pointed out in the reflective texts was the scarce involvement of some students, even in small groups. This imposed a greater burden on others: “in our group there were people who did practically nothing, making the rest of us have to do everything.” (E34).

### 3.3. Self-Directed Learning

#### 3.3.1. Facilitators

The students wrote how the autonomous work outside of class helped the groups to organize tasks and optimize time better. Thus, it increased their independence for being able to resolve doubts during the process: “there were times when you got stuck, but we helped each other, and we managed to move forward.” (P63).

In addition, the autonomous work served to develop skills for future work, which could be applied to other subjects, such as their final dissertation: “autonomous work and the independent research of the contents of the work helps in the final dissertation.” (E21). It also allowed them to use new group work tools, such as video calls, which helped them reach a consensus on the students’ individual work: “it was perfectly developed since we shared our work and doubts by video call when we were not in class.” (E74).

#### 3.3.2. Barriers

Students had difficulty working outside the classroom, as they lacked time to do so: “two of the members work and it was very difficult to meet outside of class hours to do the work.” (E54). Another barrier was negotiating and deciding the time available to carry out the work: “it was sometimes difficult to agree with all the members of the group to meet at the same time.” (E42).

Despite the careful organization, students acknowledged that sometimes, the distribution of tasks and workloads was unequal, causing some students to take on more than others: “certain people in the group worked less than the rest, meaning that others had to make up for that work.” (E7).

Finally, in their reflections, the students highlighted that the classroom sessions were insufficient and they had difficulties resolving their doubts due to the absence of the teacher during the autonomous work: “if you don’t have the support of the teacher to help clarify any doubts that arise, it’ s difficult, you hardly have time; and by email, it’s more difficult to clarify doubts compared to in person.” (E57).

## 4. Discussion

This study shows the barriers and facilitators identified in nursing students’ reflective texts regarding a teaching innovation proposal linked to the nursing research subject, which incorporated PBL, SGL, and SDL. The results obtained have been shared by previous studies in other contexts, especially in health sciences. Nonetheless, our study deepened the acquisition of research skills through a novel intervention within the subject, which simultaneously combined three active learning methodologies, unlike the studies that addressed these methodologies in isolation.

One of the novelties of our study was the application of PBL in undergraduate nursing students. In contrast, Wiggins et al. [53] and Granado-Alcón et al. [54] employed this methodology in psychology, education or medicine. Our results show that PBL facilitates greater learning, and better integration, synthesis, and consolidation of knowledge. It also facilitates the preparation of the final exams. Granado-Alcón et al. [55] describe a great acquisition of competencies and constructive learning environments through these types of methodologies. Granado-Alcón et al. [54] claim that PBL provides students with intellectual tools and improves their perception of learning. In turn, project-based approaches increase motivation [56,57], and promote attitudes toward effort, interest, and the development of skills to present assignments. Among the barriers identified in PBL, there is a lack of understanding and time to carry out the activity, together with the ambiguity of the contents. Si [58] emphasizes the challenge of dealing with “research” in students who have no previous experience, feeling perplexed and not knowing what to do. Blumenfeld et al. add that students and teachers must be prepared to successfully apply PBL [59]. Students should have basic knowledge about the study problem, adequate competencies and learning skills; meanwhile, teachers’ competencies should be oriented to teaching [59]. The lack of time to perform PBL is common in activities related to research teaching [58].

Our results on SGL describe involvement and a sense of obligation toward the task, where the work is shared, coinciding with the work by Iqbal et al. [60], in which SGL is associated with students’ commitment toward the task to be developed, enthusiasm and willingness toward the group effort. Being engaged encourages critical questioning and peer problem solving [61]. In addition, working in small groups fosters self-directed individual and group learning [62]. Our results show how the SGL enabled greater control over the assigned work, facilitated its organization and allowed the students to better understand the contributions of each member. Mennenga et al. [62] have shown that sharing points of view among team members contributes to and improves group learning. Moreover, our results showed barriers with the LMS, such as excessive workload and time commitment. Although the SGL seeks a greater involvement of students with the task, some are not sufficiently involved. This places a burden on the rest of the team and negatively impacts the group dynamics; additionally, the group effectiveness may be affected [63]. The unequal distribution of work has effects on the group, which is associated with the phenomenon of “free-rider students”. These students cause an unequal distribution of workloads, irresponsibility, and a lack of group harmony, negatively affecting the learning outcomes among students [64]. It is common for students in the group to feel that they have been treated unfairly in the face of an unequal distribution of work, especially when they all share the same situation [65]. These unfair learning situations destroy team dynamics and collaboration, which affects satisfaction with individual learning [64].

Our results show how by applying SDL, the groups organized tasks better and optimized time, increasing their independence for resolving doubts. Murad et al. describe how SDL improves skills and attitudes, as well as knowledge in the education of health science students, compared to traditional teaching methods [35]. There is a trend for the use of SDL in health professional education worldwide [66]. SDL is recommended among nursing students for the development of continuous learning skills, enabling them to achieve professional competencies [67]. This study found that one of the barriers of SDL is the lack of equity in the distribution of tasks and workloads due to the lack of time to work outside the classroom. The researchers of the present study believe that there was not enough time to fully implement SDL. Providing and reserving more time in subsequent courses would help eliminate this barrier.

This study has several limitations. Firstly, the results cannot be extrapolated; however, they can be applied to contexts with similar characteristics. Secondly, data collection was performed by means of reflective online notes; therefore, it was not possible to ask for clarifications, which reduced the amount of information obtained. Third, the learning strategies used have been classified as being of low and high efficiency according to the objectives pursued. In our case, we do not know the specific level of efficiency when the three methodologies were used in combination; therefore, it would be necessary to carry out new studies to determine this. Fourth, the impact of the use of different teaching approaches on student learning has not been measured compared with previous courses. In any case, the final grades obtained and the number of students who passed could be compared, although not the learning derived. Moreover, the previous course was taught in the context of the a pandemic, where physical presence and the follow-up of classes was disadvantaged by disruptions. Nonetheless, it would be interesting to carry out future studies to examine this issue. Finally, the students acquired knowledge using the different teaching strategies; however, we lack sufficient information related to the skills and abilities that would allow us to affirm the full acquisition of the research competencies expected of undergraduate students.

As for the practical implications for education, it is recommended that universities incorporate teaching strategies such as PBL, SGL, or SDL to facilitate the acquisition of research skills through in-depth learning. The teachers should evaluate the efficiency of these strategies after they are implemented and introduce the necessary changes to optimize learning. The use of active methodologies provides a deeper understanding, as opposed to traditional teaching–learning strategies, where students simply remember what they need to know for the exam and fail to make connections between the content taught between subjects. Moreover, the acquisition of research competences is fundamental to the students’ role as future professionals when dealing with the different problems they will face in the clinical setting. Consequently, the acquisition of research competences will enable them to respond to the conflicts they will encounter on the basis of evidence-informed practice.

We consider that the use of these approaches requires the agents involved to be familiar with them beforehand. Students must understand what to learn and teachers must know how to teach it. Lack of training and/or experience of either party could negatively influence learning objectives and outcomes. Therefore, we do not recommend the use of any of these methods by the faculty in student feedback without considering the aforementioned requirements.

## 5. Conclusions

The use of different teaching strategies in the acquisition of research competencies in nursing students was positively evaluated, and aspects to be improved were identified. In PBL, students played a more active role in their learning, were able to integrate, consolidate and synthesize the content received, put theory into practice, and better prepare for exams. In contrast, a lack of time, lack of understanding, ambiguity of content or poor guidance by teachers did not allow them to learn enough. In SGL, there was an equitable distribution of tasks, greater involvement, commitment, communication and interaction of its members, generating greater satisfaction, work environment, and work quality. Some of the negative aspects described were the workload, the dedication outside class hours, and the overload on the rest of the group when some of its members failed to become involved. In the case of SDL, tasks were better organized, time was optimized and students had a greater independence when resolving doubts. However, there was not enough time to work outside of class, a lack of equity in the distribution of time and the absence of a teacher to guide them.

The barriers encountered are relevant for improving the future teaching of the nursing research course in subsequent years. Finally, we believe that there is a great potential for the use of active methodologies to enhance nursing students’ learning processes within the subject of nursing research.

## Figures and Tables

**Figure 1 healthcare-11-01078-f001:**
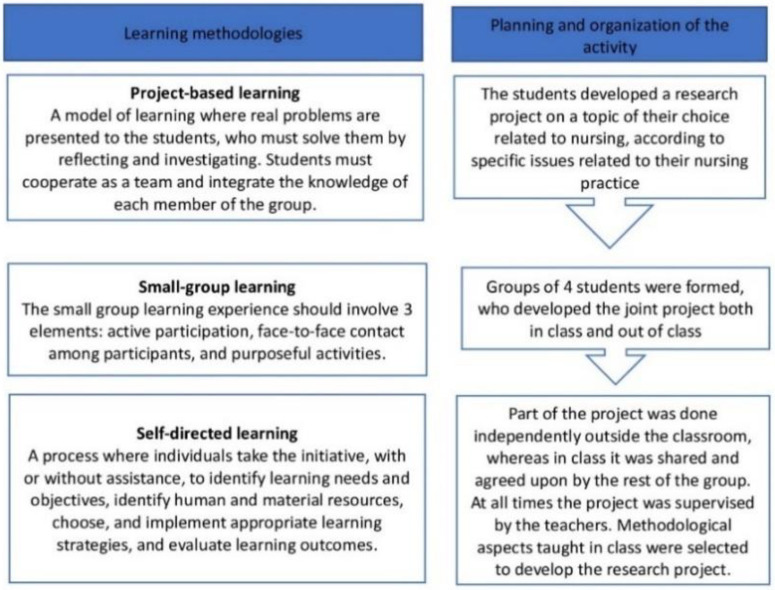
Education innovation proposal.

**Figure 2 healthcare-11-01078-f002:**
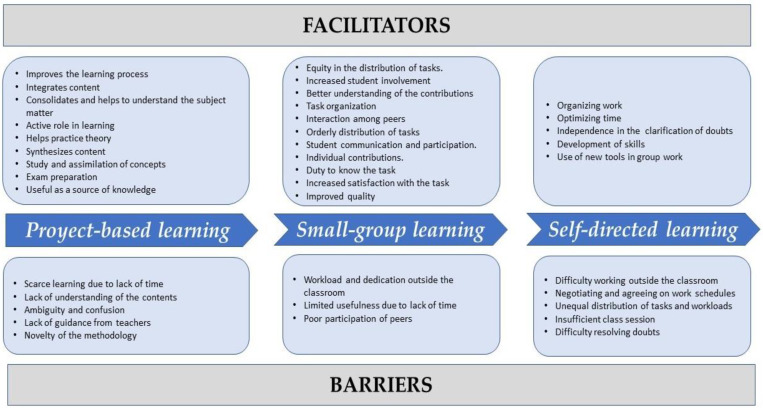
Identified facilitators and barriers.

**Table 1 healthcare-11-01078-t001:** Opening and reflexive questions.

Area of Interest	Questions
Project-based learning	What was most relevant to you about this method? How would you rate project-based learning? Was the development of the project useful for learning research content? What was the process of project-based learning like? What barriers and facilitators did you encounter?
Small-group learning	What is the most relevant aspect of this method? Did the work in small groups contribute to learning research content? How would you rate the work done in small groups (3–4 students) compared to the work done in larger groups? How was the process of working in small groups? What barriers and facilitators did you find?
Self-directed learning	What is the most relevant aspect of this method? Did the independent work contribute to learning the research content? How would you rate the work you had to do independently outside of class? How was the process of the autonomous work? What barriers and facilitators did you encounter?

## Data Availability

The data set used is locked and stored at the Nursing School of the Spanish Red Cross in Madrid, and can be obtained from the author on reasonable request.

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
