# Peer review of "Barriers and Facilitators to the Learning and Acquisition of Research Competencies among Nursing Students through Active Methodologies: A Qualitative Study Using Reflective Writing"

_healthcare, 2023, doi:10.3390/healthcare11081078_

Round 1
Reviewer 1 Report (Previous Reviewer 3)
Dear Editor,
Thank you for allowing me the opportunity to review once again the manuscript titled "Learning and Acquisition of Research Competencies in Undergraduate Nursing Students Through a Proposal for Teaching Innovation: A Qualitative Study Using Reflective Writing." I would like to congratulate the authors on their efforts to incorporate the provided feedback into the revised manuscript. I believe that the conducted peer-review process has enhanced the manuscript's scientific soundness.
Overall, I am pleased with most of the revisions made by the authors, and most elements of the COREQ recommendations were integrated into the manuscript. However, the manuscript requires some minor adjustments, including syntax and grammar revisions throughout the document. Please find below some additional comments on this newer version of the manuscript:
· On page 1, lines 40-42 (starting at "Undergraduate research was defined..."): This sentence is confusing. The authors should provide a more detailed description of how undergraduate students can be involved in research activities.
· The background section uses some concepts that are related but sometimes used in a vague way (page 2, lines 59-60). For example, competencies are the result of one's knowledge, skills, and attitudes.
· On page 2, lines 62-63, the authors state that "Knowing how to use the best available evidence enables patient care decisions to be made by integrating the most current and valid research findings, the nurse's clinical experience, values, patient preferences, and available resources." I agree with this statement, but the authors should refer to the concept of evidence-informed practice, not evidence-based practice.
· On page 2, lines 67-68 (starting with "Therefore, there is a need…"): I do not understand how this sentence relates to the one before it.
· On page 2, lines 75-78: This idea seems incomplete and would be more suitable for the discussion section.
· In your method section, please address COREQ items 17 to 23 (Data collection). Moreover, the inclusion and exclusion criteria for students are still missing.
· After carefully reviewing the discussion section, I have one last question: among the three approaches, which one would the authors recommend for less experienced teachers, taking into consideration the students' feedback?
· The last paragraph of the discussion addresses “new teaching strategies”, which is odd, since the study focuses on three very specific approaches. Please revise accordingly.
Author Response
Consulte el archivo adjunto

Reviewer 2 Report (Previous Reviewer 2)
The revised study is very appreciated as the authors have contributed more relevant knowledge about the use of Learning and acquisition of research competencies in undergraduate nursing students through a proposal for teaching innovation: A qualitative study using reflective writing. However, I have the following minor reservations that may enhance the quality of the manuscript if addressed properly:
1. The abstract is very well written and further improved.
2. Introduction and review of the literature section were short in the previous version but have now been improved by adding relevant literature.
3. A theoretical background was missing in the previous version and authors were suggested to add concrete justified theoretical roots in the literature to strengthen the manuscript’s appeal. Hence Experiential learning theory was used but not properly linked up with supportive references.
4. Don’t use the term “Previous study” rather cite the relevant reference.
5. The rationale has been justified.
6. Methodology section is very well written
7. Data analysis is very good and results are properly presented.
To sum up, the study addressed an important topic; and very good contribution to the relevant field.
Author Response
Consulte el archivo adjunto

Reviewer 3 Report (New Reviewer)
Manuscript is clearly presented and very well organized. The results and qualitative comments are balanced and illustrate the themes presented in the discussion. It was unclear to this reviewer if the faculty conducting these learning activities had received training in how to do so. I have personally used several education activities similar to these and in some I received specific training through multi-day workshops (TBL, PBL) and others I just read on my own (CBL) and the results from the students were significantly different. Some mention of faculty familiarity and experience/training with these methods should be included as this could have impacted the student responses, especially with the barriers.
Author Response
Please see the attachment

Reviewer 4 Report (New Reviewer)
Dear editor and dear authors,
Thank you for the opportunity to review your paper entitled “Learning and acquisition on research competencies in undergraduate nursing through a proposal for teaching innovation: A qualitative study using reflective writing”
The abstract was well written. Page 1, line 24, has an extra period.
The introduction, results, and discussion sections are well-framed.
The methods section needs to be clarified. The research team are the teachers who usually teach research to the nursing students.
Limitations are identified. We suggested improving the conclusions. The authors should explain to readers how these results can be applied in practice and what we should change after this research.
This study aimed to explore the perspectives of undergraduate nursing students on the barriers and facilitators after implementing a novel activity. The authors answer this question, but it will be attractive to know if the classification of these students improved compared with the others years.
In order words, what is the impact of this research?
Author Response
Please see the attachment

Reviewer 5 Report (New Reviewer)
Many thanks for the opportunity to review this study. The authors present a qualitative study exploring the barriers and facilitators following the implementation of three learning strategies within a research methodology course. The study is interesting with implications for nursing education. Please see the comments below to help strengthen the study:
1. The title of the study seems incongruent with the study aim. The key terms here are barriers and facilitators of the learning approaches, but these are missing. Kindly consider how to align these to ensure that the title is commensurate with the study objective.
2. The abstract is well raised and in line with the main study.
3. The introduction is also well raised. However, it may be helpful if the authors can provide some contextual information regarding how they deliver the research methodology course, and why a new approach is warranted.
4. The methods section is really well presented and congruent with the COREQ reporting checklist. A minor comment here will be to delete section 2.7 (quality criteria), and instead mention under the design section that the study is reported according to the COREQ checklist. Also, though the authors present a section on positionality and reflexivity, there is still a need to include a section on methodological rigor/ trustworthiness. Please consider adding this to indicate that the study was conducted rigorously.
5. Regarding ethics, I am wondering if there was any power interplay between the students and the research team? If the research team are part of the faculty delivering the course, will that in a subtle way force the students to participate? Did the students feel it was compulsory to participate?
6. It will be helpful to include the initials of who did the data collection and analysis.
7. It will be helpful if the categories and subcategories are summarised in a table to be easy for readers to navigate through.
Round 2
Reviewer 5 Report (New Reviewer)
Many thanks to the authors for thoughtfully addressing all the comments raised. I recommend acceptance of the manuscript.
Author Response
Thank you for your comment
This manuscript is a resubmission of an earlier submission. The following is a list of the peer review reports and author responses from that submission.
Round 1
Reviewer 1 Report
Dear authors,
Thank you for the opportunity to review this manuscript. I found that the authors have written the methods part well for a qualitative study. However, there was a very minimal introduction and poor breadth of literature review which demonstrated lack of gap that justify this research. The resulting themes were all already established and were very general to the teaching method rather than teaching research competencies. Hence, I suggest fresh submission where introduction is build up with local and international data, visible research gaps, results and discussion that is more critical pertaining to research competencies training.
Thank you.
Reviewer 2 Report
The research study is very appreciated as the authors have contributed significant knowledge about the use of Learning and acquisition of research competencies in undergraduate nursing students through a proposal for teaching innovation: A qualitative study using reflective writing. However, I have the following minor reservations that may enhance the quality of the manuscript if addressed properly:
1. The abstract is very well written
2. Introduction and review of the literature section are short but well written however the authors are suggested to add some recent literature on n the learning and acquisition of competencies in nursing research
3. A theoretical background is missing and authors are suggested to add concrete justified theoretical roots in the literature to strengthen the manuscript’s appeal.
4. A good rationale is developed (line number 58, 59) by the authors but need to justify their study logically.
5. Methodology section is very well written
6. Data analysis is very good and results are properly presented.
To sum up, the study addressed an important topic; and very good contribution to the relevant field.
Reviewer 3 Report
Dear Dr Rahman Shiri
Editor-in-chief of Healthcare,
I appreciate the opportunity given to review the manuscript titled “Learning and acquisition of research competencies in undergraduate nursing students through a proposal for teaching innovation: A qualitative study using reflective writing”. The authors present a qualitative exploratory study that aims to “explore the perspectives of undergraduate nursing students on the barriers and facilitators of the implementation of a teaching innovation proposal”.
After a careful review of the manuscript, I have strong reservations about its publication in the current format. Specific issues are addressed below by manuscript heading.
[Title and abstract]
- What is “educational innovation”?
- The abstract does not contain findings, since the authors just merely mentioned they have identified “facilitators and barriers to learning were identified” to the three “learning methodologies” which were not mentioned before (e.g., project-based learning; small-group learning; and self-directed learning).
- Conclusions are extremely vague.
- The keyword “research learning” does not make sense after reading the abstract. A keyword for the study design should be included.
[Introduction and study objectives]
- Page 1, line 38: Do students learn to “generate evidence-based knowledge”?
- The authors use the term “evidence-based practice” throughout their manuscript, which I have no issue with. Recent international authors and associations have proposed the term “evidence-informed practice”, given that context variables and patient preferences should also be accounted for by clinicians during decision-making.
- What do you mean by “research knowledge acquisition is scarce”? The introduction section is extremely vague and does not reference recent studies tackling this topic. Some examples are below:
https://www.sciencedirect.com/science/article/abs/pii/S1471595313001558 https://www.fons.org/library/journal/volume9-issue1/article4
https://www.scielo.br/j/reben/a/85Z5yrKyKTWsnTh8MJxqcXR/abstract/?lang=en
https://www.mdpi.com/1660-4601/17/17/6351 https://www.sciencedirect.com/science/article/abs/pii/S8755722317303629
https://www.sciencedirect.com/science/article/abs/pii/S147159531830982X
[Methods]
- Why are the authors claiming that an innovative educational intervention was implemented (consisting of PBL/SGL/SDL), when these approaches have been extensively used and reported in the literature before? I am confused about the lack of original content so far.
- Why did the authors select PBL/SGL/SDL? What evidence (with GRADE level, if possible) supports the effectiveness of each strategy?
- Page 2, line 83 – the authors state that “the researchers’ positioning was established via briefing sessions”, but fail to provide information about the positioning and previous experience in evidence-based practice, in any of the above-mentioned pedagogical strategies or even if they have regular contact with nursing students during their undergraduate training. Concerning the latter, if there is contact, how were student confidentiality and freedom of speech handled/assured by the research team?
- Inclusion/exclusion criteria is missing from subheading 2.4
- If I understood correctly, all students engaged with PBL/SGL/SDL. But I do not find any open-ended questions for students about the overall exposure to the three methodologies (e.g., were the three methodologies equally important and relevant for students? Being exposed to three different methodologies and assignments was tiring, or were students motivated throughout the course?). Furthermore, did the research team consider that exposure to the first methodology could change the students’ views on the second (and so on)?
- Table 2 is out of place.
[Discussion & Conclusion]
- The authors claim that “One of the novelties of our study was the application of PBL in undergraduate 263 nursing students”. As I stated before, PBL has been extensively referred in previous research that the authors have failed to address. I am still unsure about the study’s novelty.
- The discussion section is tangential in many aspects, mostly due to the omission of several key points that I have identified before.
- The limitations are poorly addressed by the authors.
- In light of their findings, the authors have not provided potential implications for educational practice.
- As in the abstract, the conclusions are rather vague and lack novelty.